# Training Plug-n-Play Knowledge Modules with Deep Context Distillation

## Abstract

Dynamically integrating new or rapidly evolving information after Language Model (LM) pre-training remains challenging, particularly in low-data scenarios or when dealing with private and specialized documents. In-context learning and retrieval-augmented generation (RAG) face limitations, including their high inference costs and their inability to capture global document information. In this paper, we propose a way of modularizing knowledge by training Knowledge Modules (KMs). KMs are lightweight components implemented as parameter-efficient LoRA modules, which are trained to store information about new documents and can be easily plugged into models on demand. We show that next-token prediction performs poorly in training KMs. We instead propose Deep Context Distillation: we learn KMs parameters such as to simulate hidden states and logits of a teacher that takes the document in context. Our method outperforms standard next-token prediction and pre-instruction training techniques, across two datasets. Finally, we highlight synergies between KMs and retrieval-augmented generation.

## 1 Introduction

Pre-training large language models (LLMs) on massive corpora has shown to be extremely effective at capturing a wealth of general-purpose linguistic and factual knowledge in their parameters. Adapting these models to incorporate new or rapidly evolving information remains challenging, particularly in scenarios where private or specialized documents must be integrated post-hoc. This line of research is most compelling when using LLMs in common enterprise scenarios when proprietary documents often contain the latest instructions, policies, or product details. LLMs must integrate this private data to support tasks such as question-answering (QA) or customer support; another scenario is supporting scientific discovery, where LLMs could potentially propose new hypotheses or experiments if they can ingest cutting-edge scientific publications. In both cases, we need a method that preserves the model's broad capabilities while efficiently encoding novel documents in low-data conditions. At the same time, it is useful for this knowledge to be integrated on demand, in a plug-n-play fashion.

A standard solution to these problems is *in-context learning*, wherein new, up-to-date information is provided in the context of the LLM, before the input prompt. Although many efforts have been devoted to improving long-context models, this approach hits its limits in cases where documents are extremely long. Retrieval-augmented generation (RAG) partially addresses these limitations by only selecting the most relevant passages for a given prompt (Lewis et al., 2020). However, this comes at the cost of: 1) lacking global document information, i.e. only local, or passage-level information is presented in the context, and 2) increased inference costs (in terms of memory footprint and latency) due to the context enlargement. Another alternative to capture global information is to *continually pre-train* the LLM on new documents (e.g., via parameter-efficient methods like LoRA); however, although next-token prediction is extremely effective during pre-training, it might not be effective in low-data scenarios, as recent literature suggests (Jiang et al., 2024).

In this paper, we tackle the problem of integrating information about documents in a data-efficient fashion, without knowledge of the downstream task at hand. We aim to train specialized *knowledge modules* (KMs) that encode information about the new document into continuous parameters. Encoding documents into continuous parameter is not new and traces back to Doc2Vec (Le & Mikolov, 2014) and more recently (Zhang et al., 2023). Here, we parameterize KMs as LoRA modules; this

allows for dynamically loading new knowledge on demand and aligns with recent approaches to building more modular LLMs (Ostapenko et al., 2024; Muqeeth et al., 2024). Training KMs with next-token prediction is challenging due to data scarcity. New documents contain orders of magnitude less tokens compared to pre-training. To address this problem, we enhance the learning signal through two synergistic techniques: a) knowledge distillation and b) synthetic data generation. With a), we learn the parameter of each KMs to reproduce the "behavior" of the LLM when it is presented with the new document in context. This can be seen as a generalization of Context Distillation (Snell et al., 2023). We conduct distillation from both the output probabilities and the hidden states, similar to (Sanh et al., 2019). We show that both distillation losses are important for performance. We dub this method "Deep Context Distillation" (DCD). One of the remaining questions is which inputs we should choose to elicit the behavior of the context-conditioned model. We experiment with some variants and find that synthetically generating summary data from the context-conditioned model to distill its behavior works better than the alternatives.

Concretely, our contributions are as follows:

- We introduce Knowledge Modules trained with Deep Context Distillation, which effectively condensates document knowledge into a plug-and-play parameter-efficient adapter.
- We show that DCD pairs well with synthetic data generation, thereby benefiting from additional compute.
- We evaluate our method on two long-context question answering datasets (Quality and NarrativeQA) and two base models (Phi-3 3B and Llama-3.1 8B) in two settings: open book and closed book evaluation. In all settings, DCD Knowledge Modules outperform all other methods, displaying synergies with RAG.

## 2 KNOWLEDGE MODULES

The goal of knowledge modules is to encode information about a particular document $D$ into a small set of parameters such that they can be plugged on top of a base model on demand at inference time.

### 2.1 PARAMETRIZATION

We parameterize KMs as parameter-efficient adapters using LoRA (Hu et al., 2022). For every linear layer of the base model, LoRA modifies the computation by adding the outer product of two low-rank learnable matrices $A$ and $B$. LoRA adapters have been used in modular approaches due to their ease of batching and merging (Ostapenko et al., 2024; Muqeeth et al., 2024).

### 2.2 NEXT-TOKEN PREDICTION LOSS

A straightforward way of learning a KM for a new document $D = \{d_1, \ldots, d_N\}$, where $d_i$ is a token, is to use a language modeling (LM) loss such as next-token prediction. Let's call $\kappa_D$ the parameters of the KM for document $D$, comprising $A, B$ matrices for all linear layers in the model. The LM loss trains $\kappa_D$ to be useful to predict the next token in the document:

$$\mathcal{L}_{LM} = -\sum_i \log p(d_i | d_{<i}; \kappa_D), \tag{1}$$

Many variants of this loss have been used in the past both in the original Doc2Vec (Le & Mikolov, 2014) and more recently as span prediction (Xiao et al., 2023). While next-token prediction is the de-facto approach to knowledge extraction when large amount of data is available, it suffers from the *perplexity curse* (Jiang et al., 2024), where the LLM quickly overfits and minimizes the perplexity on the next token, but fails to correctly extract all the relevant knowledge (Berglund et al., 2024).

### 2.3 DEEP CONTEXT DISTILLATION

In this work, we propose a more effective approach to learn KMs, by relying on distillation (Hinton, 2015), where the model's output is trained to match the output of a teacher network that can have access to additional information or knowledge. Our idea is based on Context Distillation (Snell et al.,

2023), originally proposed to internalize task instructions or reasoning steps into a LM. We propose to distill the behavior of a teacher model that has access to document $D$ into a student model that doesn't have access to $D$ but can optimize the parameters of the KM $\kappa_D$. In this way, we hope that $\kappa_D$ will encode the knowledge supporting powerful in-context inferences about $D$. In particular, we perform distillation both in the output (probability) space and the hidden space of the teacher LM, dubbing our loss Deep Context Distillation (DCD). A general form of DCD can be written as follows:

$$\mathcal{L}_{DCD} = \min_{\kappa_D} KL(p(\tilde{D}|D)||p(\tilde{D};\kappa_D)) + \sum_l \frac{1}{Z^l}|h^l_{\tilde{D}|D} - h^l_{\tilde{D}|\kappa}|, \qquad (2)$$

where $\tilde{D}$ denotes data that is used to compute distillation loss, $p(\tilde{D}|D)$ is the teacher, $p(\tilde{D}|\kappa_D)$ is the student that doesn't have access to $D$, $h^l$ denote the hidden states of the base LM for layer $l$ both in the teacher and the student, and $Z^l = |h^l_{\tilde{D}|D}|$ a normalization factor that depends on the L1 norm of the target hidden states. The distillation at the output layer provides a rich training signal as the full teacher distribution is available to the student, while the distillation in the hidden layer provides more direct credit assignment to every LoRA layer. A similar hidden state loss has been also used in DistilBERT (Sanh et al., 2019), albeit the authors using cosine similarity instead of L1. We found our loss to perform well in practice.

A central element left to define to make DCD work in practice is the data $\tilde{D}$ used to perform the distillation process. We decline two versions of our context-distillation loss: the first is document DCD (DDCD) and the second is summary DCD (SDCD).

**Document DCD** In DDCD, we sample random chunks of $N$ tokens from the document $D$, and use $N/2$ tokens as contextual information to give to the teacher $D$, and the remaining $N/2$ tokens as context distillation. For every position $i$ in document $D$, denote $j = i + \frac{N}{2}$ the split point; we minimize:

$$\mathcal{L}_{DCD} = \min_{\kappa_D} \sum_i KL(p(D_{i:j}|D_{j:N+i})||p(D_{i:j};\kappa_D)) + \sum_l \frac{1}{Z^l}|h^l_{D_{i:j}|D_{j:N+i}} - h^l_{D_{i:j}|\kappa}|, \quad (3)$$

**Summary DCD** In SDCD, we instead allow ourselves to generate synthetic data from the document $D$, and specifically summaries. To do so, we sample a chunk of $N$ tokens from document $D$, $D_i$ and ask the base LM to create a summary of the specific chunk, $S_i$; then we minimize:

$$\mathcal{L}_{SDCD} = \min_{\kappa_D} \sum_i KL(p(S_i|D_i)||p(S_i;\kappa_D)) + \sum_l \frac{1}{Z^l}|h^l_{S_i|D_i} - h^l_{S_i|\kappa}|, \qquad (4)$$

In practice, we create 16 summary per chunk and use every chunk-summary pair for DCD. The idea behind generating summaries is that summaries are likely to require a certain amount of inference about the information contained in the document and therefore this might in turn help encode more information into the KMs by providing a richer training signal.

## 2.4 Task Adaptation with Knowledge Extractors

The trained KM $\kappa_D$ is task-agnostic given that it is solely trained to reproduce the behavior of the teacher model when $D$ is in context. When some supervised task data is available, we might want to train additional parameters to use the knowledge stored in the KMs to maximize task performance. To do so, we train a Knowledge Extractor (KE), $\kappa_E$, a parameter-efficient module (LoRA) that can combine with the document-specific KMs to maximize task performance. For example, in the context of Question-Answering studied in this paper, if we have available a dataset of questions, answers and supporting documents $\{(q_i, a_i, D_i)\}$, we can train $\kappa_E$ to minimize the negative log-likelihood of the answers when it is combined with every document KM trained separately:

$$\mathcal{L}_{KM+KE} = \min_{\kappa_E, w} -\sum_i \log p(a_i|q_i; [\kappa_{D_i}, \kappa_E]_w), \qquad (5)$$

where $[.]$ denotes our combination function. To combine $\kappa_E$ and $\kappa_D$, we use a learnable weighted combination of the two LoRAs applied after the outer product: $[\kappa_{D_i}, \kappa_E]_w = w_M A_{D_i} B_{D_i}^T + w_E A_E B_E^T$, where $w_M, w_E$ are the weights given to the KM and KE parameters respectively. We

learn different combination parameters for every layer where LoRA is applied. We will show in the experiments that, although the KE is trained in tandem with a set of documents during training, it can generalize well to extracting knowledge from an unseen set of documents and corresponding KMs in the held-out set.

To also experiment with the synergy between KMs and retrieval-augmented generation approaches, we also extend Eq. 5, to the case where contextual information about the document is included in the context. The extension of the loss is straightforward:

$$\mathcal{L}_{RAG+KM+KE} = \min_{\kappa_E, w} - \sum_i \log p(a_i | q_i, P_1^i, \ldots, P_M^i, [\kappa_{D_i}, \kappa_E]_w), \tag{6}$$

where $P_1^i, \ldots, P_M^i$ are document passages retrieved conditioned on query $q_i$.

## 2.5 TRAINING CONSIDERATIONS

Training knowledge modules on a set of documents $\mathcal{D} = \{D_i\}$ can be done in an embarrassingly parallel fashion. In contrast, training a KE requires having the set of KMs available for joint multi-task training. KMs can be seen akin to a continuous indexing mechanism that aim to compress information about a document during training time (similar to GraphRAG (Edge et al., 2024)). Therefore, we operate with the assumption that some of the computation happening at query time can be moved into a more powerful index mechanism at training time. Training KMs requires gradient descent. In the future, efficient ways of initialize KMs to decrease computation time during indexing might be envisioned. SDCD also requires generating summaries from the document. This can be done efficiently with serving libraries such as VLLM (Kwon et al., 2023).

## 3 EXPERIMENTS

**Setup** We experiment with two question-answering datasets Quality and NarrativeQA, and two base models, Phi-3 3B and Llama-3.1 8B. Quality is a multi-choice question-answering dataset consisting of 150 training documents and 115 valid documents. The answers of the test documents are private, therefore we report the results on the dev set. Dataset have variable number of 4-way multiple-choice questions for every document. The average length of documents in the dataset is ∼5000 tokens. NarrativeQA is a question-answering dataset where documents are longer, ∼60000 tokens on average. The dataset has 1,102 training documents, 115 dev documents and 355 test documents. For each document, the question-answer pairs in the dataset are created based on a human-written summary of the full document but the evaluation is performed only on the basis of the original documents. Therefore, this dataset is especially challenging for models with limited context length as the gold answers might require assembling information across the whole document. For NarrativeQA, we evaluate performance using multi-reference Rouge-L with the gold answer, while for Quality we use Accuracy (25% being random performance).

We experiment with different setups based on the amount of information available to the models at test time. In the *closed book* setting, models do not have access to the document in context while answering the questions, therefore all the models are tested on the basis of how well they can incorporate information before solving the specific task at hand. In the *open book* setting, the document is provided in the context of the models and therefore context processing can be potentially conditioned on the question such as in RAG.

**Baselines** In closed book evaluation, we experiment with different ways of training KMs. The first is using the standard LM loss $KM_{LM}$, akin to continual pre-training. Then, we apply our document DCD loss $KM_{DDCD}$. Then, we benchmark our $KM_{SDCD}$ which uses generated summaries as target for deep context distillation. Finally, we experiment with Pre-Instruction Tuning (PIT) (Jiang et al., 2024), where they propose to concatenate task data (query/answer pairs) before the documents during continual pre-training; to be fair with our methods that do not use task data during KM training, we concatenate our generated summaries before each document and we train on the resulting concatenation. We denote this variant as $KM_{PIT}$.

In the open book setting, we use RAG and ICL as baselines: for RAG, we split each document into passages of 256 tokens and use SFR-embedding-2 (Meng et al., 2024) to embed passages and questions to perform retrieval of the top-5 relevant passages as measured by the cosine similarity with

| Phi-3 (3B) | Quality | NQA |
|---|---|---|
| *Closed Book* | | |
| Base Model | 26.6 | 12.1 |
| $KM_{LM}$ | 26.4 | 15.2 |
| $KM_{DDCD}$ | 28.7 | 21.2 |
| $KM_{PIT}$ | 29.4 | 11.5 |
| $KM_{SDCD}$ | **32.5** | **23.9** |
| $KM_{LM}$ + KE | 36.0 | 20.7 |
| $KM_{DDCD}$ + KE | 40.4 | 28.0 |
| $KM_{SDCD}$ + KE | **47.1** | **32.2** |
| *Open Book* | | |
| ICL | 33.1 | 21.0 |
| RAG | 34.7 | 23.0 |
| RAG + $KM_{SDCD}$ | **35.1** | **23.1** |
| RAG + KE | 53.4 | 36.2 |
| RAG + $KM_{SDCD}$ + KE | **55.8** | **39.1** |

| Llama3.1 (8B) | Quality | NQA |
|---|---|---|
| *Closed Book* | | |
| Base Model | 12.5 | 20.9 |
| $KM_{LM}$ | 26.1 | 17.9 |
| $KM_{SDCD}$ | **37.2** | **26.6** |
| $KM_{LM}$ + KE | 40.9 | 22.3 |
| $KM_{SDCD}$ + KE | **57.2** | **32.5** |
| *Open Book* | | |
| ICL | 46.0 | 38.0 |
| RAG | **41.0** | **28.2** |
| RAG + $KM_{SDCD}$ | 40.1 | 27.5 |
| RAG + KE | 62.2 | 36.7 |
| RAG + $KM_{SDCD}$ + KE | **64.1** | **39.7** |

Table 1: Results for Quality and NarrativeQA and on Phi-3 3B (left) and Llama3.1 8B (right).

each question. Similarly to KM, we assume we know the document the questions relate to and therefore we only retrieve passages from that document (we don't search over the set of all documents in the dataset). We report results of zero-shot RAG (just denoted as RAG) and a fine-tuned version of RAG (RAG + KE), where a KE module (just a LoRA adapter) is fine-tuned on the task with the RAG context. For all methods, KEs are always trained solely on the training documents, and never on the dev/test documents. We experiment with combinations of RAG and KMs, RAG + KM (zero-shot) and RAG + KE + KM (KE trained to combine RAG and KM information) to analyze the synergy between RAG and KMs. Technically, ICL for decoder-only models can be considered as a closed book approach if the KV document cache is stored in memory. However, this comes at an excessive storage cost (for Llama3.1 8B, for a document of 60k tokens, it's ~30Gb). We experiment with KV compression methods such as the recently proposed L2 compress (Devoto et al., 2024b) to analyze tradeoffs between performance and storage cost.

KMs and KEs methods use LoRA as parameter-efficient adapter with a rank of 16, lora alpha of 16, learning rate of 1e-3, are trained with 1500 steps with batch size of 8 and cosine learning rate scheduler. We use greedy decoding to sample the responses for NarrativeQA.

**Results** We report the results for the two base models and the two datasets in Table 1. Results are consistent and show that SDCD performs best across the board in the closed book setting, outperforming both LM, PIT and DDCD. In the open book setting, we see that KMs struggle at zero-shot combination with RAG (RAG + KM lines in both table). For LLama-3.1, there are slight signs of forgetting (compare RAG vs. RAG + KM), which might be due to the fact that KMs are never trained to cope with long contexts. However, these might be readily solved by training a specialized KE, which highlight strongly synergistic behavior of RAG and KMS (compare RAG + KE and RAG + KM + KE; +3.4% R-L and +3% R-L in NQA and ~2% Acc. on Quality).

In Figure 1, we relate performance on NQA vs. token cost at inference time, as measured as the number of context tokens the model consume to produce an answer. We denote by $k$ the number of retrieved passages for RAG. We see that KM+KE (without any retrieval) outperform retrieving RAG+KE with $k = 1$ while only use the question tokens in the prompt (40 tokens in average vs 200 tokens for RAG+KE with $k = 1$). Scaling the number of retrieved passages $k$ benefit both RAG + KM + KE and RAG + KE, while introducing KM retains gains over RAG for similar values of $k$ and matches performance at a lower cost (42.4 obtained with $k = 8$ vs 42.5 attained with $k = 16$) providing savings of 50% at inference time.

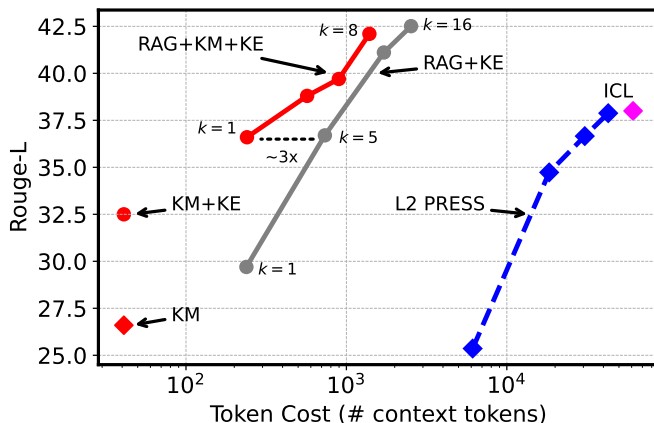

Figure 1: Number of tokens in the context for every model (on the x-axis) vs Rouge-L performance (on the y-axis) on the NarrativeQA dataset. We report both zero-shot and (diamond marker) and fine-tuned models (circle marker, KE). We also benchmark a recent KV-cache compression method based on the l2 norm of the key vectors (Devoto et al., 2024b).

## 4 RELATED WORK

**In-Context Learning and RAG** have been used to incorporate external knowledge at inference time. While powerful, these techniques can impose high storage costs (e.g., storing large key-value caches). Efforts to compress KV caches typically use heuristics such as norms of the key-value pairs (Devoto et al., 2024a) or learned. Recent methods try to build more powerful RAG mechanisms by scaling both indexing and generation (Yue et al., 2024; Edge et al., 2024). Yue et al. (2024) extend RAG by retrieving passages, queries, and answers. They further scale inference time by decomposes queries into sub-queries, incrementally building solutions. GraphRAG Edge et al. (2024) has a costly indexing step which builds a knowledge graph from chunked documents by extracting (subject, predicate, object) relations, then clusters and summarizes sub-graphs. This offline graph-based pre-processing is reminiscent of KMs in reducing repeated computation at inference time. Much like KMs, GraphRAG help capturing global document context. Data generated by GraphRAG at indexing time could potentially be used as distillation data in KMs.

**Knowledge Distillation** typically transfers logits or intermediate representations from a teacher model to a student (Xu et al., 2024). While effective at compressing large models, these approaches usually differ from KMs in that they do not explicitly store domain knowledge for repeated use in specific tasks. Context distillation (Snell et al., 2023) aims to "absorb" a context into a model in a way that preserves its impact on generation. Early work focused on toxicity and safety contexts (Askell et al., 2021) and for internalizing reasoning traces (Snell et al., 2023). Recently, Shin et al. (2024) propose Generative Context Distillation, augmenting the distillation objective with a generative loss to match model outputs more comprehensively.

**Knowledge Injection with Modules** Several works have similar goals of KMs. Xiao et al. (2023) introduce "plug-and-play" document modules where an MLP transforms encoded document tokens into soft prompts used across attention layers. Zhang et al. (2023) similarly train knowledge modules in a task-agnostic or task-specific manner, caching the processed document representation for efficient reuse. Amortization-based approaches train a "hypernetwork" to generate lightweight adaptations (e.g., low-rank shifts or prefix-tuning parameters) from a given context. Chen et al. (2024) learn to project context tokens into low-rank parameter shifts in a self-supervised manner, and Tack et al. (2024) encode documents into prefix-tuning weights using a T5-based hypernetwork. These methods train the hypernet on multi-task data, so at inference time it can produce task-specific or document-specific modules in a single forward pass. KMs as described in this paper are trained with gradient descent on single documents independent of any multi-task training. This increases per-document costs but reduces the risk of domain shift. Future work might be devoted to efficient learning of KMs.

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
