# OpenReview forum: "Training Plug n' Play Knowledge Modules with Deep Context Distillation"
_ICLR.cc/2025/Workshop/MCDC — MCDC @ ICLR 2025_

### Official Review · Reviewer_Ygxj · 2025-02-27

**Rating:** 7
**Confidence:** 3
**Fit:** 4

**Summary:**

The authors propose a novel method of modularising document knowledge in Language Models using LoRA adapters. They name these lightweight components as Knowledge Modules (KM) and train them using Deep Context Distillation, an alternative training procedure to next-token prediction. Their experiments show improved performance over baseline conditions under different settings.

**Reason For Giving A Higher Score:**

The concept is novel, with great experimental results, and fits the theme of using modular components to maximize the potential of language models with plug-and-play fine-tuned adapters.

**Reason For Giving A Lower Score:**

One limitation is lack of transferability between different models, even with the same architecture, since each KM has to be trained to adapt to the model's pre-trained weights.

**Strengths And Weaknesses:**

Strengths:
1. A novel, lightweight design of modular knowledge storage that is plug-and-play and so can be easy to use when adapting a language model to understand different document knowledge without re-training and storing multiple models.
2. Interesting method of training these KMs using deep context distillation, it is an intuitive method since we want the model to have a comprehend the entire document and its specific context. While next token prediction mainly trains the model to be able to understand the structure of language in general.
3. Compatible with standard methods like RAG and has significant performance improvements in the closed book setting and when combining with RAG for open book.

Weaknesses:
1. KMs do not seem transferable to different models, and each set of KMs must be re-trained to fit each model.
2. In the closed book setting, KMs do not seem to perform too well on its own, even when combined with RAG, but requires an additional training step with Knowledge Extractors (KEs).

**Suggestions:**

1. In document DCD, does the teacher predict the first N/2 tokens from the last N/2 tokens? (Eqn. 3) perhaps the other way around would be better? or using a masked token prediction for the teacher.
2. Could it be possible to train a KM for the open book setting?
3. Is it possible to use other adapters besides LoRA?

Minor: typos on line 127, 138, table 1 (right), under open book, should ICL be bolded instead?

---

### Official Review · Reviewer_FuH9 · 2025-02-28

**Rating:** 5
**Confidence:** 3
**Fit:** 4

**Summary:**

In this paper authors propose a way of modularizing knowledge by training Knowledge Modules (KMs) which are trained to store information about new documents and can be easily plugged into pre-trained models on demand. Authors claim that this method serves to be more effective than the traditional approaches of RAG and ICL specially in cases when documents are too long leading to high inference cost. Although I agree with the premise and the problem described, I am not fully convinced about the novelty of solution presented in the paper which seems marginally incremental to existing works on knowledge distillation and more recently context distillation, both of which are referred to in the paper. Similarly, the results presented on open-book setting aren't particularly impressive which calls for additional evidence needed in terms of ablation-- what's the added benefit of Knowledge Extractors (KEs) on top of KMs? Although the results in Table 1show that KE+KM+RAG is superior, I'd like the authors to dive deeper into explaining it further.

**Reason For Giving A Higher Score:**

N/A

**Reason For Giving A Lower Score:**

The paper has potential but in its current form doesn't meet the bar from my perspective. My concerns are around the lack of ablation studies, the importance of KMs, KMs+RAG on top of KEs. More experiments and intuitive explanation on why these components improve the KEs will help address the concerns.

**Strengths And Weaknesses:**

Strength: The problem discussed in the paper fits well with the theme of the workshop and is a real one as approaches like RAG and ICL do hit their limits pretty soon when the documents are long or the memory of sequence-calling of LLMs gets bloated. The paper is reasonably well written and I was able to easily follow it for the most part.


Weakness: The contribution doesn't meet the bar. The proposed way of training KMs is adopted from works like Context Distillation using a combination of KL loss and L1 loss on hidden states. The notion of backpropagating the hidden states loss was demonstrated by Sanh et al., 2019. The notable change made was switching the cosine loss to an L1 loss. Furthermore, I am not fully convinced of the value added by KEs on top of KMs. More ablation studies could help here.

**Suggestions:**

The paper has potential but in its current form doesn't meet the bar from my perspective. My concerns are around the lack of ablation studies, the importance of KMs, KMs+RAG on top of KEs. More experiments and intuitive explanation on why these components improve the KEs will help address the concerns.

---

### Official Review · Reviewer_8VhE · 2025-03-03

**Rating:** 6
**Confidence:** 3
**Fit:** 3

**Summary:**

This paper proposes a method for efficiently integrating knowledge from new or specialized documents into LLMs by training plug-and-play Knowledge Modules. KMs are lightweight LoRA-based modules trained using DCD to emulate the behavior of LLMs within a document context, thereby encoding knowledge into parameters without requiring access to the document. DCD combines output probability and hidden state distillation while leveraging synthetic summary data to enhance learning signals. Experiments show that KMs outperform traditional next-token prediction and pre-instruction training methods in both closed-book and open-book question-answering tasks.

**Reason For Giving A Higher Score:**

See weaknesses. In particular, the proposed method relies on a teacher model with the same architecture that already contains the knowledge from new documents. This raises the question of whether distilling a separate student model is truly necessary.

**Reason For Giving A Lower Score:**

The idea of incorporating document knowledge through LoRA is highly intriguing.

**Strengths And Weaknesses:**

Strengths:
* The idea of leveraging LoRA as a Knowledge Module to embed knowledge from new documents is interesting.
* The experimental results demonstrate the effectiveness of the proposed method.
Weaknesses:
* Training LoRA to encode document knowledge requires additional training, which can be costly in scenarios with frequent knowledge updates.
* Finding a suitable teacher model that contains new knowledge is challenging, especially when the teacher and student models must share the same architecture. In such cases, it may be more efficient to use the teacher model directly for downstream tasks rather than distilling a student model from it.

**Suggestions:**

* Provide a more in-depth discussion of the second weakness to better illustrate the necessity of the proposed method.
If multiple documents need to be incorporated, consider constructing a Multi-LoRA system for knowledge management.
* Adding relevant literature and discussions, such as [1-3], could strengthen the argument.

[1] LoraRetriever: Input-Aware LoRA Retrieval and Composition for Mixed Tasks in the Wild

[2] A Survey on Model MoErging: Recycling and Routing Among Specialized Experts for Collaborative Learning

[3] Towards Modular LLMs by Building and Reusing a Library of LoRAs

---

### Decision · Program_Chairs · 2025-03-06

**Decision:**

Accept

**Comment:**

This paper is a good fit for the workshop and has been positively received by all the reviewers. We encourage the authors to take reviewers' comments and suggestions into consideration, especially ablations proposed by FuH9, for the final version of the paper.